# $\ell_1$-regression with Heavy-tailed Distributions

**Lijun Zhang,    Zhi-Hua Zhou**
National Key Laboratory for Novel Software Technology
Nanjing University, Nanjing 210023, China
{zhanglj, zhouzh}@lamda.nju.edu.cn

## Abstract

In this paper, we consider the problem of linear regression with heavy-tailed distributions. Different from previous studies that use the squared loss to measure the performance, we choose the absolute loss, which is capable of estimating the conditional median. To address the challenge that both the input and output could be heavy-tailed, we propose a truncated minimization problem, and demonstrate that it enjoys an $\widetilde{O}(\sqrt{d/n})$ excess risk, where $d$ is the dimensionality and $n$ is the number of samples. Compared with traditional work on $\ell_1$-regression, the main advantage of our result is that we achieve a high-probability risk bound without exponential moment conditions on the input and output. Furthermore, if the input is bounded, we show that the classical empirical risk minimization is competent for $\ell_1$-regression even when the output is heavy-tailed.

## 1  Introduction

Linear regression used to be a mainstay of statistics, and remains one of our most important tools for data analysis [Hastie et al., 2009]. Let $\mathcal{T} = \{(\mathbf{x}_1, y_1), \ldots, (\mathbf{x}_n, y_n)\} \subseteq \mathbb{R}^d \times \mathbb{R}$ be a set of input-output pairs that are independently drawn from an unknown distribution $\mathbb{P}$. In linear regression, we assume that the relationship between the input and output can be well modeled by a linear function, and aim to discover it from the training set $\mathcal{T}$. For a linear function $f(\mathbf{x}) = \mathbf{x}^\top \mathbf{w}$, its quality is measured by the expected prediction error on a random pair $(\mathbf{x}, y)$ sampled from $\mathbb{P}$, i.e., the *risk*:

$$R_\ell(\mathbf{w}) = \mathrm{E}_{(\mathbf{x},y)\sim\mathbb{P}} \left[ \ell(\mathbf{x}^\top \mathbf{w}, y) \right]$$

where $\ell(\cdot, \cdot)$ is a loss that quantifies the prediction error. The most popular losses include the squared loss $\ell_2(u, v) = (u - v)^2$ and the absolute loss $\ell_1(u, v) = |u - v|$.

Let $\mathcal{W} \subseteq \mathbb{R}^d$ be a domain of linear coefficients. The standard approach for linear regression is the empirical risk minimization (ERM)

$$\min_{\mathbf{w}\in\mathcal{W}} \widehat{R}_\ell(\mathbf{w}) = \frac{1}{n} \sum_{i=1}^{n} \ell(\mathbf{x}_i^\top \mathbf{w}, y_i)$$

which selects the linear function that minimizes the empirical risk on the training set. In the literature, there are plenty of theoretical guarantees on linear regression by ERM, either targeting linear regression directly [Birgé and Massart, 1998, Györfi et al., 2002], or being derived from the general theories of statistical learning [Vapnik, 2000, Koltchinskii, 2011, Zhang et al., 2017]. In order to establish high-probability risk bounds, most of the previous analyses rely on the assumption that the input and output are bounded or sub-Gaussian.

However, if the input and output are heavy-tailed, which is commonly encountered in many disciplines such as finance and environment [Finkenstädt and Rootzén, 2003], existing high-probability bounds of ERM become invalid. In fact, for heavy-tailed distributions, it has been explicitly proved that the

empirical risk is no longer a good approximation of the risk [Catoni, 2012], which inspires recent studies on learning with heavy-tailed losses [Audibert and Catoni, 2011, Hsu and Sabato, 2014, Brownlees et al., 2015]. In particular, for linear regression with squared loss, i.e., $\ell_2$-regression, Audibert and Catoni [2011] develop a truncated min-max estimator, and establish an $O(d/n)$ excess risk that holds with high probability even when the input and output are heavy-tailed. This result is a great breakthrough as it extends the scope of linear regression, but is limited to the squared loss.

The motivation of squared loss is to estimate the conditional mean of $y$ given $\mathbf{x}$. Besides the squared loss, there also exist other losses for regression. For example, if we are interested in the conditional median, then absolute loss would be a natural choice [Hastie et al., 2009]. Furthermore, absolute loss is more robust in that it is resistant to outliers in the data [Peter J. Rousseeuw, 1987]. In this paper, we target linear regression with absolute loss, namely $\ell_1$-regression. Inspired by the truncation function of Audibert and Catoni [2011], we propose a truncated minimization problem to support heavy-tailed distributions. Theoretical analysis shows that our method achieves an $\widetilde{O}(\sqrt{d/n})$ excess risk, which holds with high probability. Our theoretical guarantee requires very mild assumptions, and is derived from standard techniques—the covering number and concentration inequalities. Furthermore, we show that if the input is bounded, the classical ERM is sufficient even when the output is heavy-tailed.

We highlight the contributions of this paper as follows.

- We propose a truncated minimization problem for $\ell_1$-regression, which is more simple than the truncated min-max formulation of Audibert and Catoni [2011] for $\ell_2$-regression.
- This is the first time an $\widetilde{O}(\sqrt{d/n})$ excess risk is established for $\ell_1$-regression under the condition that both the input and output could be heavy-tailed. Although Brownlees et al. [2015] develop a general theorem for heavy-tailed losses, when applied to $\ell_1$-regression, it requires the input to be bounded.
- When the input is bounded, we prove that the classical ERM achieves an $O(D/\sqrt{n})$ excess risk for $\ell_1$-regression, where $D$ is the maximum norm of the input, and does not require any assumption on the output.

Finally, we note that although this paper focuses on $\ell_1$-regression, the idea of truncated minimization and our analysis can be directly extended to Lipschitz losses [Alquier et al., 2017] that satisfy

$$|\ell(\mathbf{x}^\top \mathbf{w}, y) - \ell(\mathbf{x}^\top \mathbf{w}', y)| \leq |\mathbf{x}^\top \mathbf{w} - \mathbf{x}^\top \mathbf{w}'|, \ \forall \mathbf{w}, \mathbf{w}' \in \mathcal{W}.$$

Many popular losses for classification, including hinge loss and logistic loss, are Lipschitz losses. We will provide detailed investigations in an extended paper.

## 2 Related Work

In this section, we review the recent work on learning with heavy-tailed losses. A distribution $F$ is heavy-tailed if and only if [Foss et al., 2013]

$$\int e^{\lambda x} dF(x) = \infty, \ \forall \lambda > 0.$$

There exist studies that try to equip ERM with theoretical guarantees under heavy-tailed distributions. However, these theories need additional assumptions, such as the small-ball assumption [Mendelson, 2014, 2015] and the Bernstein's condition [Dinh et al., 2016, Alquier et al., 2017], and do not always hold with high probability. In the following, we only discuss alternatives of ERM.

### 2.1 Truncation based Approaches

Our truncated method follows the line of research stemmed from Catoni [2012]. In his seminal work, Catoni [2012] considers the estimation of the mean and variance of a real random variable from independent and identically distributed (i.i.d.) samples. Let $x_1, \ldots, x_n \in \mathbb{R}$ be i.i.d. samples drawn from some unknown probability distribution $\mathbb{P}$. Catoni [2012] builds the estimator $\widehat{\theta}$ of the mean as the solution of the equation

$$\sum_{i=1}^{n} \psi \left[ \alpha(x_i - \widehat{\theta}) \right] = 0 \tag{1}$$

where $\alpha > 0$ is a positive parameter, and $\psi(\cdot) : \mathbb{R} \mapsto \mathbb{R}$ is a truncation function that is non-decreasing and satisfies

$$- \log \left( 1 - x + \frac{x^2}{2} \right) \leq \psi(x) \leq \log \left( 1 + x + \frac{x^2}{2} \right). \tag{2}$$

By choosing $\alpha = \sqrt{2/n\nu}$, where $\nu$ is the variance of the random variable, Catoni [2012] demonstrates that with high probability, the deviation of $\widehat{\theta}$ from the mean is upper bounded by $O(\sqrt{\nu/n})$. Thus, (1) can be applied to heavy-tailed distributions, as long as the variance is finite.

Audibert and Catoni [2011] extend the robust estimator of Catoni [2012] to linear regression with squared loss. Specifically, they consider the $\ell_2$-norm regularized $\ell_2$-regression (i.e., ridge regression), and propose the following min-max estimator

$$\min_{\mathbf{w} \in \mathcal{W}} \max_{\mathbf{u} \in \mathcal{W}} \lambda \left( \|\mathbf{w}\|^2 - \|\mathbf{u}\|^2 \right) + \frac{1}{\alpha n} \sum_{i=1}^{n} \psi \left[ \alpha(y_i - \mathbf{x}_i^\top \mathbf{w})^2 - \alpha(y_i - \mathbf{x}_i^\top \mathbf{u})^2 \right] \tag{3}$$

where $\| \cdot \|$ denotes the $\ell_2$-norm of vectors and $\psi(\cdot)$ is the truncation function defined as

$$\psi(x) = \begin{cases} - \log \left( 1 - x + \frac{x^2}{2} \right), 0 \leq x \leq 1; \\ \qquad \log(2), x \geq 1; \\ \qquad -\psi(-x), x \leq 0. \end{cases} \tag{4}$$

Let $(\widehat{\mathbf{w}}, \widehat{\mathbf{u}})$ be the optimal solution to (3). With a suitable choice of $\alpha$, Audibert and Catoni [2011] have proved that the following risk bound

$$R_{\ell_2}(\widehat{\mathbf{w}}) + \lambda \|\widehat{\mathbf{w}}\|^2 - \min_{\mathbf{w} \in \mathcal{W}} \left\{ R_{\ell_2}(\mathbf{w}) + \lambda \|\mathbf{w}\|^2 \right\} = O \left( \frac{d}{n} \right)$$

holds with high probability even when neither the input nor the output has exponential moments.

In a subsequent work, Brownlees et al. [2015] apply the robust estimator of Catoni [2012] to the general problem of learning with heavy-tailed losses. Let $\mathbf{x}$ be a random variable taking values in some measurable space $\mathcal{X}$ and $\mathcal{F}$ be a set of nonnegative functions defined on $\mathcal{X}$. Given $n$ independent random variables $\mathbf{x}_1, \ldots, \mathbf{x}_n$, all distributed as $\mathbf{x}$, Brownlees et al. [2015] propose to use the robust estimator in (1) to estimate the mean of each function $f \in \mathcal{F}$, and choose the one that minimizes the estimator. Formally, the optimization problem is given by

$$\min_{f \in \mathcal{F}} \quad \widehat{\mu}_f$$

$$\text{s. t.} \quad \frac{1}{n\alpha} \sum_{i=1}^{n} \psi \left[ \alpha(f(\mathbf{x}_i) - \widehat{\mu}_f) \right] = 0$$

where $\psi(\cdot)$ is defined as

$$\psi(x) = \begin{cases} \log \left( 1 + x + \frac{x^2}{2} \right), x \geq 0; \\ - \log \left( 1 - x + \frac{x^2}{2} \right), x \leq 0. \end{cases} \tag{5}$$

Based on the generic chaining method [Talagrand, 2005], Brownlees et al. [2015] develop performance bounds of the above problem. However, the theoretical guarantees rely on the condition that the function space $\mathcal{F}$ is bounded in terms of certain distance. When applying their results to $\ell_1$-regression, the linear function needs to be bounded, which means the input vector must be bounded [Brownlees et al., 2015, Section 4.1.1]. When applying to $\ell_2$-regression, the input also needs to be bounded and the theoretical guarantee is no longer a high-probability bound [Brownlees et al., 2015, Section 4.1.2].

## 2.2 Median-of-means Approaches

Another way to deal with heavy-tailed distributions is the median-of-means estimator [Nemirovski and Yudin, 1983, Alon et al., 1999, Bubeck et al., 2013, Minsker, 2015]. The basic idea is to divide

the data into several groups, calculate the sample mean within each group, and take the median of these means. Recently, Hsu and Sabato [2014, 2016] generalize the median-of-means estimator to arbitrary metric spaces, and apply it to the minimization of smooth and strongly convex losses. Specifically, for $\ell_2$-regression with heavy-tailed distributions, a high-probability $\widetilde{O}(d/n)$ excess risk is established, under slightly stronger assumptions than those of Audibert and Catoni [2011]. For regression problem, Lugosi and Mendelson [2016] introduce a new procedure, the so-called median-of-means tournament, which achieves the optimal tradeoff between accuracy and confidence under minimal assumptions. The setting of Lugosi and Mendelson [2016] is general in the sense that the function space could be any convex class of functions, not necessary linear, but the performance is only measured by the squared loss.

Compared with truncation based approaches, the advantage of median-of-means approaches is that they do not require prior knowledge of distributional properties. However, the current theoretical results of median-of-means are restricted to the squared loss or strongly convex losses, and thus cannot be applied to $\ell_1$-regression considered in this paper.

## 3 Our Results

In this section, we first present our truncated minimization problem, then discuss its theoretical guarantee, and finally study the special setting of bounded inputs.

### 3.1 Our Formulation

Inspired by the truncated min-max estimator of Audibert and Catoni [2011], we propose the following truncated minimization problem for $\ell_1$-regression with heavy-tailed distributions:

$$\min_{\mathbf{w} \in \mathcal{W}} \frac{1}{n\alpha} \sum_{i=1}^{n} \psi\big(\alpha|y_i - \mathbf{x}_i^\top \mathbf{w}|\big) \tag{6}$$

where the truncation function $\psi(\cdot)$ is non-decreasing and satisfies (2), and $\alpha > 0$ is a parameter whose value will be determined later. Note that we can choose (4) or (5) as the truncation function.

Compared with the min-max problem in (3), our minimization problem in (6) is more simple. Although (6) is still a non-convex problem, it has a special structure that can be exploited. Because $\psi(\cdot)$ is non-decreasing, it is easy to verify that each individual function $\psi(\alpha|y_i - \mathbf{x}_i^\top \mathbf{w}|)$ is quasiconvex. Thus, our problem is to minimize the sum of quasiconvex functions. From the recent developments of quasiconvex optimization [Hazan et al., 2015], we may apply (stochastic) normalized gradient descent (NGD) to solve (6). This paper focuses on the statistical property of (6), and we leave the design of efficient optimization procedures as a future work.

### 3.2 Theoretical Guarantees

Let $\mathcal{W}$ be a subset of a Hilbert space $\mathcal{H}$, and $\|\cdot\|$ be the norm associated with the inner product of $\mathcal{H}$. We introduce assumptions that used in our analysis.

**Assumption 1** *The domain $\mathcal{W}$ is totally bounded such that for any $\varepsilon > 0$, there exists a finite $\varepsilon$-net of $\mathcal{W}$.*[1]

**Assumption 2** *The expectation of the squared norm of $\mathbf{x}$ is bounded, that is,*

$$\mathrm{E}_{(\mathbf{x},y)\sim\mathbb{P}}\big[\|\mathbf{x}\|^2\big] < \infty.$$

**Assumption 3** *The $\ell_2$-risk of all $\mathbf{w} \in \mathcal{W}$ is bounded, that is,*

$$\sup_{\mathbf{w}\in\mathcal{W}} R_{\ell_2}(\mathbf{w}) = \sup_{\mathbf{w}\in\mathcal{W}} \mathrm{E}_{(\mathbf{x},y)\sim\mathbb{P}}\big[(y - \mathbf{x}^\top \mathbf{w})^2\big] < \infty.$$

**Remark 1** We have the following comments regarding our assumptions.

- Although Assumption 1 requires the domain $\mathcal{W}$ is bounded, the input and output could be unbounded, which allows us to model heavy-tailed distributions.
- Because our goal is to bound the $\ell_1$-risk, which is the first-order moment, it is natural to require higher-order moment conditions. Thus, in Assumptions 2 and 3, we introduce second-order moment conditions on inputs and outputs. Our assumptions support heavy-tailed distributions in the sense that commonly used heavy-tailed distributions, such as the Pareto distribution (with parameter $\alpha > 2$) and the log-normal distribution, have finite second-order moment.
- By Jensen's inequality, we have $(\mathrm{E}[\|\mathbf{x}\|])^2 \leq \mathrm{E}[\|\mathbf{x}\|^2]$. Thus, Assumption 2 implies $\mathrm{E}[\|\mathbf{x}\|]$ is bounded.
- Given Assumptions 1 and 2, Assumption 3 can be relaxed as the $\ell_2$-risk of the optimal solution $\mathbf{w}_*$ is bounded. To see this, we have

$$
\begin{aligned}
R_{\ell_2}(\mathbf{w}) \leq & 2R_{\ell_2}(\mathbf{w}_*) + 2(\mathbf{w} - \mathbf{w}_*)^\top \mathrm{E}\left[\mathbf{x}\mathbf{x}^\top\right](\mathbf{w} - \mathbf{w}_*) \\
\leq & 2R_{\ell_2}(\mathbf{w}_*) + 2\|\mathbf{w} - \mathbf{w}_*\|^2 \left\|\mathrm{E}\left[\mathbf{x}\mathbf{x}^\top\right]\right\|_2
\end{aligned}
$$

where $\|\cdot\|_2$ is the spectral norm of matrices. First, Assumption 1 implies $\|\mathbf{w} - \mathbf{w}_*\|$ is bounded. Second, Assumption 2 implies the spectral norm of $\mathrm{E}[\mathbf{x}\mathbf{x}^\top]$ is also bounded, that is,

$$
\left\|\mathrm{E}\left[\mathbf{x}\mathbf{x}^\top\right]\right\|_2 \leq \mathrm{E}\left[\|\mathbf{x}\mathbf{x}^\top\|_2\right] = \mathrm{E}\left[\|\mathbf{x}\|^2\right] < \infty.
$$

Thus, $R_{\ell_2}(\mathbf{w})$ is bounded as long as $R_{\ell_2}(\mathbf{w}_*)$ is bounded and Assumptions 1 and 2 hold.

To present our theoretical guarantee, we introduce the notation of the covering number. The minimal cardinality of the $\varepsilon$-net of $\mathcal{W}$ is called the covering number and denoted by $N(\mathcal{W}, \varepsilon)$. Let $\widehat{\mathbf{w}}$ be a solution to (6), and $\mathbf{w}_* \in \operatorname{argmin}_{\mathbf{w} \in \mathcal{W}} R_{\ell_1}(\mathbf{w})$ be an optimal solution that minimizes the $\ell_1$-risk. We have the following excess risk bound.

**Theorem 1** *Let $0 < \delta < 1/2$. Under Assumptions 1, 2 and 3, with probability at least $1 - 2\delta$, we have*

$$
R_{\ell_1}(\widehat{\mathbf{w}}) - R_{\ell_1}(\mathbf{w}_*) \leq 2\varepsilon\mathrm{E}[\|\mathbf{x}\|] + \alpha\varepsilon^2\mathrm{E}\left[\|\mathbf{x}\|^2\right] + \frac{3\alpha}{2}\sup_{\mathbf{w} \in \mathcal{W}} R_{\ell_2}(\mathbf{w}) + \frac{1}{n\alpha}\log\frac{N(\mathcal{W}, \varepsilon)}{\delta^2}
$$

*for any $\varepsilon > 0$. Furthermore, by setting*

$$
\alpha = \sqrt{\frac{1}{n}\log\frac{N(\mathcal{W}, \varepsilon)}{\delta^2}},
$$

*we have*

$$
R_{\ell_1}(\widehat{\mathbf{w}}) - R_{\ell_1}(\mathbf{w}_*) \leq 2\varepsilon\mathrm{E}[\|\mathbf{x}\|] + \sqrt{\frac{1}{n}\log\frac{N(\mathcal{W}, \varepsilon)}{\delta^2}}\left(\varepsilon^2\mathrm{E}\left[\|\mathbf{x}\|^2\right] + \frac{3}{2}\sup_{\mathbf{w} \in \mathcal{W}} R_{\ell_2}(\mathbf{w}) + 1\right).
$$

**Remark 2** Note that Theorem 1 is very general in the sense that it can be applied to *infinite* dimensional Hilbert spaces, provided the domain $\mathcal{W}$ has a finite covering number [Cucker and Smale, 2002]. In contrast, the result of Audibert and Catoni [2011] is limited to finite spaces. The difference is caused by the different techniques used in the analysis: While Audibert and Catoni [2011] employ the PAC-Bayesian analysis, we make use of the covering number and standard concentrations.

To reveal the order of the excess risk, we need to specify the value of the covering number. To this end, we consider the special case that $\mathcal{W}$ is a bounded subset of Euclidean space, and introduce the following condition.

**Assumption 4** *The domain $\mathcal{W}$ is a subset of $\mathbb{R}^d$ and its radius is bounded by $B$, that is,*

$$
\|\mathbf{w}\| \leq B, \ \forall \mathbf{w} \in \mathcal{W} \subseteq \mathbb{R}^d. \tag{7}
$$

Let $\mathcal{B}_r \subseteq \mathbb{R}^d$ be a ball centered at origin with radius $r$, and $\mathcal{N}(\mathcal{B}_r, \varepsilon)$ be its $\varepsilon$-net with minimal cardinality, denoted by $N(\mathcal{B}_r, \varepsilon)$. According to a standard volume comparison argument [Pisier, 1989], we have

$$
\log N(\mathcal{B}_1, \varepsilon) \leq d\log\frac{3}{\varepsilon} \Rightarrow \log N(\mathcal{B}_r, \varepsilon) \leq d\log\frac{3r}{\varepsilon}.
$$

Since $\mathcal{W} \subseteq \mathcal{B}_B$, we have

$$\log N(\mathcal{W}, \varepsilon) \leq \log N\left(\mathcal{B}_B, \frac{\varepsilon}{2}\right) \leq d \log \frac{6B}{\varepsilon}$$

where the first inequality is because the covering numbers are (almost) increasing by inclusion [Plan and Vershynin, 2013, (3.2)]. Then, we have the following corollary by setting $\varepsilon = 1/n$.

**Corollary 2** *Let $0 < \delta < 1/2$, and set*

$$\alpha = \sqrt{\frac{1}{n} \log \frac{N(\mathcal{W}, 1/n)}{\delta^2}}.$$

*Under Assumptions 2, 3 and 4, with probability at least $1 - 2\delta$, we have*

$$R_{\ell_1}(\widehat{\mathbf{w}}) - R_{\ell_1}(\mathbf{w}_*)$$

$$\leq \frac{2}{n} \mathrm{E}[\|\mathbf{x}\|] + \sqrt{\frac{1}{n} \left( d \log(6nB) + \log \frac{1}{\delta^2} \right) \left( \frac{1}{n^2} \mathrm{E}\left[\|\mathbf{x}\|^2\right] + \frac{3}{2} \sup_{\mathbf{w} \in \mathcal{W}} R_{\ell_2}(\mathbf{w}) + 1 \right)}$$

$$= O\left( \sqrt{\frac{d \log n}{n}} \right).$$

**Remark 3** Ignoring the logarithmic factor, Corollary 2 shows an $\widetilde{O}(\sqrt{d/n})$ excess risk that holds with high probability. From the above discussions, we see that the square root dependence on $d$ comes from the upper bound of the covering number. If the domain $\mathcal{W}$ has additional structures (e.g., sparse), the covering number may have a smaller dependence on $d$, and as a result, the dependence on $d$ could be improved.

## 3.3 Bounded Inputs

If we only allow the output to be heavy-tailed and the input is bounded, the problem becomes much easier. Although both our method and the algorithm of Brownlees et al. [2015] are applicable, at least in theory, there is no need to resort to sophisticated methods. In fact, a careful analysis shows that the classical ERM is sufficient in this case.

We introduce the following assumption.

**Assumption 5** *The norm of the random vector $\mathbf{x} \in \mathbb{R}^d$ is upper bounded by a constant $D$, that is,*

$$\|\mathbf{x}\| \leq D, \ \forall (\mathbf{x}, y) \sim \mathbb{P}. \tag{8}$$

Then, we have the following risk bound for ERM.

**Theorem 3** *Let*

$$\bar{\mathbf{w}} \in \underset{\mathbf{w} \in \mathcal{W}}{\operatorname{argmin}} \frac{1}{n} \sum_{i=1}^{n} |y_i - \mathbf{x}_i^{\top} \mathbf{w}|$$

*be a solution returned by ERM. Under Assumptions 4 and 5, with probability at least $1 - \delta$, we have*

$$R_{\ell_1}(\bar{\mathbf{w}}) - R_{\ell_1}(\mathbf{w}_*) \leq \frac{4BD}{\sqrt{n}} \left( 1 + \sqrt{\frac{1}{2} \log \frac{1}{\delta}} \right).$$

**Remark 4** When inputs are upper bounded, we do not need any assumption about outputs. That is because the absolute loss is 1-Lipschitz continuous, and outputs will be canceled in the analysis. Theorem 3 implies ERM achieves an $O(D/\sqrt{n})$ excess risk which holds with high probability. Compared with the risk bound in Corollary 2, we observe that the new bound is independent from the dimensionality $d$, but it has a linear dependence on $D$, which is the upper bound of the norm of inputs.

## 4 Analysis

Due to the limitation of space, we only present the proof of Theorem 1. The proof of Theorem 3 can be found in the full paper [Zhang and Zhou, 2018].

## 4.1 Proof of Theorem 1

To simplify notations, define

$$\widehat{R}_{\psi \circ \ell_1}(\mathbf{w}) = \frac{1}{n\alpha} \sum_{i=1}^{n} \psi\big(\alpha|y_i - \mathbf{x}_i^\top \mathbf{w}|\big).$$

From the optimality of $\widehat{\mathbf{w}}$, we have

$$\underbrace{\frac{1}{n\alpha} \sum_{i=1}^{n} \psi\big(\alpha|y_i - \mathbf{x}_i^\top \widehat{\mathbf{w}}|\big)}_{\widehat{R}_{\psi \circ \ell_1}(\widehat{\mathbf{w}})} \leq \underbrace{\frac{1}{n\alpha} \sum_{i=1}^{n} \psi\big(\alpha|y_i - \mathbf{x}_i^\top \mathbf{w}_*|\big)}_{\widehat{R}_{\psi \circ \ell_1}(\mathbf{w}_*)}. \tag{9}$$

Next, we will discuss how to upper bound $\widehat{R}_{\psi \circ \ell_1}(\mathbf{w}_*)$ by $R_{\ell_1}(\mathbf{w}_*)$ and lower bound $\widehat{R}_{\psi \circ \ell_1}(\widehat{\mathbf{w}})$ by $R_{\ell_1}(\widehat{\mathbf{w}})$.

Because $\mathbf{w}_*$ is independent from samples $(\mathbf{x}_1, y_1), \ldots, (\mathbf{x}_n, y_n)$, it is easy to relate $\widehat{R}_{\psi \circ \ell_1}(\mathbf{w}_*)$ with $R_{\ell_1}(\mathbf{w}_*)$ by the standard concentration techniques. To this end, we have the following lemma.

**Lemma 1** *With probability at least $1 - \delta$, we have*

$$\widehat{R}_{\psi \circ \ell_1}(\mathbf{w}_*) \leq R_{\ell_1}(\mathbf{w}_*) + \frac{\alpha}{2} R_{\ell_2}(\mathbf{w}_*) + \frac{1}{n\alpha} \log \frac{1}{\delta}.$$

Lower bounding $\widehat{R}_{\psi \circ \ell_1}(\widehat{\mathbf{w}})$ is more involved because $\widehat{\mathbf{w}}$ depends on the sample. To this end, we combine the covering number and concentration inequalities to develop the following lemma.

**Lemma 2** *With probability at least $1 - \delta$, we have*

$$R_{\ell_1}(\widehat{\mathbf{w}}) \leq \widehat{R}_{\psi \circ \ell_1}(\widehat{\mathbf{w}}) + 2\varepsilon \mathrm{E}[\|\mathbf{x}\|] + \alpha \sup_{\mathbf{w} \in \mathcal{W}} R_{\ell_2}(\mathbf{w}) + \alpha \varepsilon^2 \mathrm{E}\left[\|\mathbf{x}\|^2\right] + \frac{1}{n\alpha} \log \frac{N(\mathcal{W}, \varepsilon)}{\delta}$$

*for any $\varepsilon > 0$.*

Then, Theorem 1 is a direct consequence of (9), Lemmas 1 and 2, and the union bound.

## 4.2 Proof of Lemma 1

First, note that our truncation function $\psi$ satisfies

$$\psi(x) \leq \log\left(1 + x + \frac{x^2}{2}\right), \quad \forall x \in \mathbb{R}. \tag{10}$$

Then, we have

$$\begin{aligned}
\mathrm{E}\left[\exp\left(n\alpha \widehat{R}_{\psi \circ \ell_1}(\mathbf{w}_*)\right)\right] &= \mathrm{E}\left[\exp\left(\sum_{i=1}^{n} \psi\big(\alpha|y_i - \mathbf{x}_i^\top \mathbf{w}_*|\big)\right)\right] \\
&\overset{(10)}{\leq} \mathrm{E}\left[\prod_{i=1}^{n}\left(1 + \alpha|y_i - \mathbf{x}_i^\top \mathbf{w}_*| + \frac{\alpha^2(y_i - \mathbf{x}_i^\top \mathbf{w}_*)^2}{2}\right)\right] \\
&= \left(\mathrm{E}\left[1 + \alpha|y - \mathbf{x}^\top \mathbf{w}_*| + \frac{\alpha^2|y - \mathbf{x}^\top \mathbf{w}_*|^2}{2}\right]\right)^n \\
&= \left(1 + \alpha R_{\ell_1}(\mathbf{w}_*) + \frac{\alpha^2}{2} R_{\ell_2}(\mathbf{w}_*)\right)^n \\
&\overset{1+x \leq e^x}{\leq} \exp\left(n\left[\alpha R_{\ell_1}(\mathbf{w}_*) + \frac{\alpha^2}{2} R_{\ell_2}(\mathbf{w}_*)\right]\right).
\end{aligned} \tag{11}$$

By Chernoff's method [Lugosi, 2009], we have

$$
\begin{aligned}
&\mathrm{P}\left\{n\alpha\widehat{R}_{\psi\circ\ell_1}(\mathbf{w}_*) \geq n\left[\alpha R_{\ell_1}(\mathbf{w}_*) + \frac{\alpha^2}{2}R_{\ell_2}(\mathbf{w}_*)\right] + \log\frac{1}{\delta}\right\} \\
&= \mathrm{P}\left\{\exp\left(n\alpha\widehat{R}_{\psi\circ\ell_1}(\mathbf{w}_*)\right) \geq \exp\left(n\left[\alpha R_{\ell_1}(\mathbf{w}_*) + \frac{\alpha^2}{2}R_{\ell_2}(\mathbf{w}_*)\right] + \log\frac{1}{\delta}\right)\right\} \\
&\leq \frac{\mathrm{E}\left[\exp\left(n\alpha\widehat{R}_{\psi\circ\ell_1}(\mathbf{w}_*)\right)\right]}{\exp\left(n\left[\alpha R_{\ell_1}(\mathbf{w}_*) + \frac{\alpha^2}{2}R_{\ell_2}(\mathbf{w}_*)\right] + \log\frac{1}{\delta}\right)} \overset{(11)}{\leq} \delta
\end{aligned}
$$

which completes the proof.

### 4.3 Proof of Lemma 2

Let $\mathcal{N}(\mathcal{W},\varepsilon)$ be an $\varepsilon$-net of $\mathcal{W}$ with minimal cardinality $N(\mathcal{W},\varepsilon)$. From the definition of $\varepsilon$-net, there must exist a $\widetilde{\mathbf{w}} \in \mathcal{N}(\mathcal{W},\varepsilon)$ such that $\|\widehat{\mathbf{w}} - \widetilde{\mathbf{w}}\| \leq \varepsilon$. So, we have

$$
|y_i - \mathbf{x}_i^\top\widehat{\mathbf{w}}| \geq |y_i - \mathbf{x}_i^\top\widetilde{\mathbf{w}}| - |\mathbf{x}_i^\top(\widetilde{\mathbf{w}} - \widehat{\mathbf{w}})| \geq |y_i - \mathbf{x}_i^\top\widetilde{\mathbf{w}}| - \varepsilon\|\mathbf{x}_i\|. \tag{12}
$$

Since $\psi(\cdot)$ is non-decreasing, we have

$$
\widehat{R}_{\psi\circ\ell_1}(\widehat{\mathbf{w}}) = \frac{1}{n\alpha}\sum_{i=1}^n \psi\left(\alpha|y_i - \mathbf{x}_i^\top\widehat{\mathbf{w}}|\right) \overset{(12)}{\geq} \frac{1}{n\alpha}\sum_{i=1}^n \psi\left(\alpha|y_i - \mathbf{x}_i^\top\widetilde{\mathbf{w}}| - \alpha\varepsilon\|\mathbf{x}_i\|\right). \tag{13}
$$

To proceed, we develop the following lemma to lower bound the last term in (13).

**Lemma 3** *With probability at least $1 - \delta$, for all $\widetilde{\mathbf{w}} \in \mathcal{N}(\mathcal{W},\varepsilon)$, we have*

$$
\begin{aligned}
&\frac{1}{n\alpha}\sum_{i=1}^n \psi\left(\alpha|y_i - \mathbf{x}_i^\top\widetilde{\mathbf{w}}| - \alpha\varepsilon\|\mathbf{x}_i\|\right) \\
&\geq R_{\ell_1}(\widetilde{\mathbf{w}}) - \left[\varepsilon\mathrm{E}[\|\mathbf{x}\|] + \alpha\sup_{\mathbf{w}\in\mathcal{W}} R_{\ell_2}(\mathbf{w}) + \alpha\varepsilon^2\mathrm{E}\left[\|\mathbf{x}\|^2\right] + \frac{1}{n\alpha}\log\frac{N(\mathcal{W},\varepsilon)}{\delta}\right].
\end{aligned} \tag{14}
$$

Substituting (14) into (13), with probability at least $1 - \delta$, we have

$$
\begin{aligned}
&\widehat{R}_{\psi\circ\ell_1}(\widehat{\mathbf{w}}) \\
&\geq R_{\ell_1}(\widetilde{\mathbf{w}}) - \left[\varepsilon\mathrm{E}[\|\mathbf{x}\|] + \alpha\sup_{\mathbf{w}\in\mathcal{W}} R_{\ell_2}(\mathbf{w}) + \alpha\varepsilon^2\mathrm{E}\left[\|\mathbf{x}\|^2\right] + \frac{1}{n\alpha}\log\frac{N(\mathcal{W},\varepsilon)}{\delta}\right] \\
&\geq R_{\ell_1}(\widehat{\mathbf{w}}) - \left[2\varepsilon\mathrm{E}[\|\mathbf{x}\|] + \alpha\sup_{\mathbf{w}\in\mathcal{W}} R_{\ell_2}(\mathbf{w}) + \alpha\varepsilon^2\mathrm{E}\left[\|\mathbf{x}\|^2\right] + \frac{1}{n\alpha}\log\frac{N(\mathcal{W},\varepsilon)}{\delta}\right]
\end{aligned}
$$

where the last step is due to the following inequality

$$
R_{\ell_1}(\widehat{\mathbf{w}}) = \mathrm{E}\left[|y - \mathbf{x}^\top\widehat{\mathbf{w}}|\right] \leq \mathrm{E}\left[|y - \mathbf{x}^\top\widetilde{\mathbf{w}}| + |\mathbf{x}^\top(\widetilde{\mathbf{w}} - \widehat{\mathbf{w}})|\right] \leq R_{\ell_1}(\widetilde{\mathbf{w}}) + \varepsilon\mathrm{E}[\|\mathbf{x}\|].
$$

### 4.4 Proof of Lemma 3

We first consider a fixed $\widetilde{\mathbf{w}} \in \mathcal{N}(\mathcal{W},\varepsilon) \subseteq \mathcal{W}$. The proof is similar to that of Lemma 1. Recall that the truncation function $\psi(\cdot)$ satisfies

$$
\psi(x) \geq -\log\left(1 - x + \frac{x^2}{2}\right), \quad \forall x \in \mathbb{R}. \tag{15}
$$

Then, we have

$$
\mathrm{E}\left[\exp\left(-\sum_{i=1}^{n}\psi\big(\alpha|y_i-\mathbf{x}_i^\top\widetilde{\mathbf{w}}|-\alpha\varepsilon\|\mathbf{x}_i\|\big)\right)\right]
$$

$$
\overset{(15)}{\leq}\mathrm{E}\left[\prod_{i=1}^{n}\left(1-\alpha|y_i-\mathbf{x}_i^\top\widetilde{\mathbf{w}}|+\alpha\varepsilon\|\mathbf{x}_i\|+\frac{\alpha^2\left(|y_i-\mathbf{x}_i^\top\widetilde{\mathbf{w}}|-\varepsilon\|\mathbf{x}_i\|\right)^2}{2}\right)\right]
$$

$$
=\left(\mathrm{E}\left[1-\alpha|y-\mathbf{x}^\top\widetilde{\mathbf{w}}|+\alpha\varepsilon\|\mathbf{x}\|+\frac{\alpha^2\left(|y-\mathbf{x}^\top\widetilde{\mathbf{w}}|-\varepsilon\|\mathbf{x}\|\right)^2}{2}\right]\right)^n \qquad (16)
$$

$$
=\left(1-\alpha R_{\ell_1}(\widetilde{\mathbf{w}})+\alpha\varepsilon\mathrm{E}[\|\mathbf{x}\|]+\frac{\alpha^2}{2}\mathrm{E}\left[\left(|y-\mathbf{x}^\top\widetilde{\mathbf{w}}|-\varepsilon\|\mathbf{x}\|\right)^2\right]\right)^n
$$

$$
\leq\exp\left[n\Big(-\alpha R_{\ell_1}(\widetilde{\mathbf{w}})+\alpha\varepsilon\mathrm{E}[\|\mathbf{x}\|]+\alpha^2 R_{\ell_2}(\widetilde{\mathbf{w}})+\alpha^2\varepsilon^2\mathrm{E}\left[\|\mathbf{x}\|^2\right]\Big)\right].
$$

where the last step is due to the basic inequalities $1+x\leq e^x$ and $(a+b)^2\leq 2a^2+2b^2$.

By Chernoff's method [Lugosi, 2009], we have

$$
\mathrm{P}\left\{-\sum_{i=1}^{n}\psi\big(\alpha|y_i-\mathbf{x}_i^\top\widetilde{\mathbf{w}}|-\alpha\varepsilon\|\mathbf{x}_i\|\big)\geq\right.
$$

$$
\left. n\Big(-\alpha R_{\ell_1}(\widetilde{\mathbf{w}})+\alpha\varepsilon\mathrm{E}[\|\mathbf{x}\|]+\alpha^2 R_{\ell_2}(\widetilde{\mathbf{w}})+\alpha^2\varepsilon^2\mathrm{E}\left[\|\mathbf{x}\|^2\right]\Big)+\log\frac{1}{\delta}\right\}
$$

$$
=\mathrm{P}\left\{\exp\left(-\sum_{i=1}^{n}\psi\big(\alpha|y_i-\mathbf{x}_i^\top\widetilde{\mathbf{w}}|-\alpha\varepsilon\|\mathbf{x}_i\|\big)\right)\geq\right.
$$

$$
\left.\exp\left[n\Big(-\alpha R_{\ell_1}(\widetilde{\mathbf{w}})+\alpha\varepsilon\mathrm{E}[\|\mathbf{x}\|]+\alpha^2 R_{\ell_2}(\widetilde{\mathbf{w}})+\alpha^2\varepsilon^2\mathrm{E}\left[\|\mathbf{x}\|^2\right]\Big)+\log\frac{1}{\delta}\right]\right\}
$$

$$
\leq\frac{\mathrm{E}\left[\exp\left(-\sum_{i=1}^{n}\psi\big(\alpha|y_i-\mathbf{x}_i^\top\widetilde{\mathbf{w}}|-\alpha\varepsilon\|\mathbf{x}_i\|\big)\right)\right]}{\exp\left[n\Big(-\alpha R_{\ell_1}(\widetilde{\mathbf{w}})+\alpha\varepsilon\mathrm{E}[\|\mathbf{x}\|]+\alpha^2 R_{\ell_2}(\widetilde{\mathbf{w}})+\alpha^2\varepsilon^2\mathrm{E}\left[\|\mathbf{x}\|^2\right]\Big)+\log\frac{1}{\delta}\right]}\overset{(16)}{\leq}\delta.
$$

Thus, with probability at least $1-\delta$, we have

$$
-\frac{1}{n\alpha}\sum_{i=1}^{n}\psi\big(\alpha|y_i-\mathbf{x}_i^\top\widetilde{\mathbf{w}}|-\alpha\varepsilon\|\mathbf{x}_i\|\big)
$$

$$
\leq -R_{\ell_1}(\widetilde{\mathbf{w}})+\varepsilon\mathrm{E}[\|\mathbf{x}\|]+\alpha R_{\ell_2}(\widetilde{\mathbf{w}})+\alpha\varepsilon^2\mathrm{E}\left[\|\mathbf{x}\|^2\right]+\frac{1}{n\alpha}\log\frac{1}{\delta}
$$

$$
\leq -R_{\ell_1}(\widetilde{\mathbf{w}})+\varepsilon\mathrm{E}[\|\mathbf{x}\|]+\alpha\sup_{\mathbf{w}\in\mathcal{W}}R_{\ell_2}(\mathbf{w})+\alpha\varepsilon^2\mathrm{E}\left[\|\mathbf{x}\|^2\right]+\frac{1}{n\alpha}\log\frac{1}{\delta}.
$$

We complete the proof by taking the union bound over all $\widetilde{\mathbf{w}}\in\mathcal{N}(\mathcal{W},\varepsilon)$.

## 5 Conclusion and Future Work

In this paper, we consider $\ell_1$-regression with heavy-tailed distributions, and propose a truncated minimization problem. Under mild assumptions, we prove that our method enjoys an $\widetilde{O}(\sqrt{d/n})$ excess risk, which holds with high probability. Compared with traditional work on $\ell_1$-regression, the main advantage of our result is that we establish a high-probability bound without exponential moment conditions on the input and output. Furthermore, we demonstrate that when the input is bounded, the classical ERM is sufficient for $\ell_1$-regression.

In the future, we will develop optimization algorithms and theories for the non-convex problem in (6). Another future work is to apply the idea of truncated minimization to other losses in machine learning, especially Lipschitz losses.

**Acknowledgments**

This work was partially supported by the NSFC (61751306), YESS (2017QNRC001), and the Collaborative Innovation Center of Novel Software Technology and Industrialization. We thank an anonymous reviewer of COLT 2018 for helping us simplify the proof of Theorem 1.

## Footnotes

[1] A subset $\mathcal{N} \subseteq \mathcal{K}$ is called an $\varepsilon$-net of $\mathcal{K}$ if for every $\mathbf{w} \in \mathcal{K}$ one can find a $\widetilde{\mathbf{w}} \in \mathcal{N}$ so that $\|\mathbf{w} - \widetilde{\mathbf{w}}\| \leq \varepsilon$.

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
