[Reviews · NeurIPS 2018]

Reviewer 1



In this manuscript the authors consider the problem of linear regression in the case where the distributions of the input and output are heavy-tailed. The goal of this task is to find a linear function that is gives the best description for transforming the input into the observed output. Here the term best means that this function should minimize the risk, which is the expected difference between the function applied to the input and the observed output under a specific loss function. Although in many standard work on linear regression the squared loss is considered, the authors argue that for heavy-tailed in- and output the absolute loss is more suited. To approach the linear regression problem the authors use a truncation method, which finds a linear function that minimizes a rescaled empirical average of a so-called truncation function. The difference between the risk of the true linear regression and that of the truncation method is called the excess risk. The main results of this manuscript provide bounds on the excess risk, under certain conditions on the input distribution. In particular, using epsilon nets, they obtain an improved excess risk bound for linear regression with absolute loss, which they further improve using additional assumptions on the input data. The overall setup of the manuscript is clear. The authors explain the problem of linear regression and comment on different approaches and techniques in the literature, especially related to heavy-tailed distributions. The proofs are easy to follow and I did not find any errors. The problem of linear regression with heavy-tailed distributions for input and even output data is relevant since many interesting systems in the real-world exhibit such behaviors. The innovative aspect of this work is the combination of absolute loss linear regression and truncation methods for a wide class of truncation functions. My main concern with this work is that although the title and many parts of the text refer to heavy-tailed distributions, these are never defined. I understand that one has an intuitive idea of what heavy-tailed means, but these are well defined objects, see for instance Definition 2.2 or Theorem 2.6 in Foss et al. An introduction to Heavy-tailed and subexponential distributions. The reason I bring this up is that nowhere in any of the proofs could I find any argument that used properties of heavy-tailed distributions. All results in this manuscript are proven under the three main assumptions. It is also worth noting that often real-world data is heavy-tailed with diverging second moment, which violates Assumption 2. Moreover, it is not clear at all if one should expect Assumption 3 to hold for simple heavy-tailed distributions such as Pareto or Log-normal. This is crucial since if this is not the case then it would make the results stated here much less useful. To summarize, the paper is well-written and addresses a relevant problem by making a novel combination of absolute loss linear regression with truncation methods. The proofs are easy to read and no errors were found in them. However, the manuscript does not make use of any properties of heavy-tailed distributions and does not even discuss the validity of its assumptions for some standard examples of such distributions. Therefore, I would only weakly recommend this paper for acceptance to NIPS 2018. ADDED AFTER FEEDBACK: The authors addressed my questions and concerns regarding the use of the term heavy-tailed and the existence of second moments. Although I personally still believe that working with heavy-tailed (more specifically regularly-varying) distributions in the setting of infinite second moment seems more realistic, I see the authors point of view and am fine with the setting used in this paper. My other small questions where also addressed properly and hence I stand by my previous decision on this paper. Below is a small list of additional comments. Main text: Line 29, [heavy-tailed]: Please provide a definition of this. Equation (2): Why do you need the \frac{x^2}{2} term inside the function. It seems that this forces the finite second moment condition? Could you elaborate a bit on this? Line 77: Here you seem to indicate that a paper from 2011 extends a result from 2012. I did not find any information that the 2012 was available before the one from 2011. Could you maybe rephrase this sentence. Assumption 2: How realistic is the finite second moment assumption with respect to real-world data? Especially in the heavy-tailed regime this might be false most of the times. Could you discuss this a bit? Line 143-144: I do not understand why you would not simply use this relaxed assumption. Could you explain this? Line 201, [the omitted ones…]: What does ones refer to here. This is not really clear from the text. Are they lemmas or equations?

Reviewer 2



The paper deals with l_1 regression when the noise is only assumed to have 2 moments. To proceed, the authors minimize an estimation of the l_1 loss constructed using the estimators of univariate means by Catoni (2012). Using basic concentration inequalities in an approach similar to Vapnik, the authors derive rates of convergence for this estimator in sqrt(d log n/n). I think that it would be interesting to have a deeper investigation of the Gaussian case to have a fair benchmark to compare these rates. The strategy developed for the l_2 case follows these lines and it would be interesting to see if the minmax strategy that was necessary in the l_2 case to obtain optimal rates is also necessary in the l_1 case or if the min estimator proposed here is sufficient to get optimal rates. As the results are stated, it seems that there is gap between rates of the ERM in the bounded case and the rates obtained for the robust estimators. It suggests that the rates in the robust case are not optimal and it would be nice to see if the difference between these rates is necessary.

Reviewer 3



Summary: The authors study the problem of regression when the loss function is \ell_1 and the noise is heavy tailed. They provide results for a truncated estimator when the input and output are unbounded. Pros: 1. The proofs are well written and are easy to follow. 2. The authors do a thorough literature survey and do well to motivate the problem and place their work in comparison to past work on this problem. Cons: 1. The estimator proposed is highly non-convex and it is not clear if it is a computationally tractable in practice for any problems of interest. 2. Corollary 2 is misleading in its claims. It assumes that \sup_{w \in W} R_{\ell_2}(w) is bounded by a constant. This is not true even when the noise process is Gaussian. For example if the set W is the unit ball in \ell_2 and noise added is Gaussian, then the term \sup_{w \in W} R_{\ell_2}(w) will scale as O(d) making their result d^{3/2}/n. Apart from this the results of Theorem 1 fail to capture the scaling of the sample complexity with the variance of the noise process. It feels to me that this scaling is hidden away in terms like \sup_{w} R_{l_2}(w) which are harder to interpret. 3. All of the proofs following by simple standard arguments and it is not clear to me where the technical novelty lies in this paper. Their proposal of the estimator is new (although modified from Audibert and Cantoni). However as previously mentioned this estimator is computationally intractable. 4. Theorem 3 appear to be (very) easily obtained from classical results such as those in Bartlett & Mendelson 2002. I do not see what is novel in this result. Suggestions: 1. Adding empirical results will be very useful I think to convince readers that this estimator is tractable. Currently this estimator appears to be non-convex and hard to optimize. It would also help to verify the scaling obtained theoretically. 2. An analysis that eschews bounding the excess risk in terms of maximum ell_2 risk would also be insightful to reveal the scaling of the sample complexity with the properties of the noise process. Minor Comments: 1. On Ln. 160 Converging number -> Covering number 2. On Ln. 209 by the standard technique of concentrations -> by standard concentration techniques.

Reviewer 4



Paper summary: This paper studies linear regression with the absolute loss. The goal is to develop estimators that work well when the moments of the data beyond the second-order could possibly not exist. This is the "heavy tail" situation studied in many previous works, but not for linear regression with absolute loss. In this paper, an estimator based on solving a non-convex optimization problem is proposed and analyzed. The optimization problem uses Catoni's truncation technique, which has been used in many previous works on learning under "heavy tail" situations. A risk bound is developed for the estimator. Another setting where the covariates are also bounded is studied, and it is shown that empirical risk minimization succeeds in that case. Review: The estimator proposed is very similar to one proposed by Brownlees, Joly, and Lugosi in their 2015 paper. In the context of regression with absolute loss, the analysis of Brownlees et al require the prediction functions be $L_\infty$-bounded. So the contribution of the present paper is, for linear regression, a modification of the estimator and accompanying analysis that work when the covariates could be unbounded. Technically, in the present paper, the Catoni risk estimate $\mu$ used in the Brownlees et al objective is (optimistically) dropped from the objective function. This is a simple idea that works out nicely, using fairly simple/standard techniques. I checked the proofs of the main result; they seem to be correct.) Throughout the paper, the authors make some rather inappropriate comparisons between $\ell_1$- and $\ell_2$-regression, because the goals are rather different. The absolute loss is motivated with the goal of estimating conditional median. The squared loss is motivated with the goal of estimating conditional mean. If robustness was the main concern, but the goal was to estimate conditional mean, then absolute loss would not really be appropriate unless the noise was symmetric. So I think the authors should use some care when discussing the absolute loss and why it is used, and also when making comparisons to other works that are motivated by estimating conditional mean. The total boundedness of the predictor domain $\mathcal{W}$ is a restriction not present in some of the previous work. It is similar to regularization. The analysis by Audibert and Catoni was very interesting because (among other reasons) it could be applied to ordinary least squares, where there is no regularization. Theorem 3 is essentially a special case of standard Rademacher complexity results for Lipschitz losses and linear predictors. The only subtlety is that the loss is not bounded, but difference of losses (relative to any fixed predictor) are, under the assumption of bounded $x$'s and bounded $w$'s. Anyway, that is only needed for the concentration part of the generalization bound; the Rademacher part is already known. Note that for $\ell_2$-regression, the same techniques would not apply (and instead one must use smoothness and other properties). Anyway, I don't think this is a major contribution of the paper, but it was nice that the authors pointed it out. The final comment regarding Lipschitz losses seems to come too late! I think throughout reading the paper, it was clear that the Lipschitz-property of the loss would be the key to making the estimator work. The focus on $\ell_1$-regression felt a bit odd, partly because of the aforementioned issues regarding motivation. I think the paper would have felt stronger if the details would have been worked out for Lipschitz losses. Overall: - The main result is interesting and new. - There is a nice idea in the paper. - The paper is a bit "light" in terms of the strength of the main result, and the other claimed major result is a bit of an oversell. - The paper needs a major revision in terms of how the problem and results are discussed. More comments: - Display below line 79: I suggest you clarify what norm is being used here (and in the rest of the paper). Also, I think there is an extraneous $|$ around there. - Line 160: Typo, "converging numbers". - Remark 3: This is an inappropriate comparison, as discussed above. I suggest either clarifying this point or removing the remark. - Line 176: Typo, "$\epsilon$". - Corollary 2 and Remark 4: The big-$O$ used here is hiding some moments. I think if you only care about the rate, then $d$ need not appear; if you care about other things beyond the rate, then you should also care about these moments.